# Analysis and Design of an X-Band Reflectarray Antenna for Remote Sensing Satellite System

**DOI:** 10.3390/s22031166

**Published:** 2022-02-03

**Authors:** Shimaa A. M. Soliman, Eman M. Eldesouki, Ahmed M. Attiya

**Affiliations:** Microwave Engineering Department, Electronics Research Institute, Joseph Tito St., Huckstep, El Nozha, Cairo 11843, Egypt; eman@eri.sci.eg (E.M.E.); attiya@eri.sci.eg (A.M.A.)

**Keywords:** reflectarray antenna, flat-top radiation pattern, remote sensing satellite system, genetic algorithm

## Abstract

This paper presents the analysis and design of an X-band reflectarray. The proposed antenna can be used for a medium Earth orbit (MEO) remote sensing satellite system in the 8.5 GHz band. To obtain a nearly constant response along the coverage area of this satellite system, the proposed antenna was designed with a flat-top radiation pattern with a beam width of around 29° for the required MEO system. In addition, broadside pencil beam and tilted pencil beam reflectarrays were also investigated. The feeding element of the proposed reflectarray antennas is a Yagi–Uda array. The amplitude and phase distribution of the fields due to the feeding element on the aperture of the reflectarray antenna are obtained directly by numerical simulation without introducing any approximation. The required phase distribution along the aperture of the reflectarray to obtain the required flat-top radiation pattern is obtained using the genetic algorithm (GA) optimization method. The reflecting elements of the reflectarray are composed of stacked circular patches. This stacked configuration was found to be appropriate for obtaining a wide range of reflection phase shift, which is required to implement the required phase distribution on the reflectarray aperture. The antenna was fabricated and measured for verification.

## 1. Introduction

Recently, satellite communication systems in low and medium Earth orbit (LEO/MEO) have experienced rapid development. Satellites are characterized by their design-and-deployment cost, power consumption, and down-link bandwidth [1]. In order to increase the down-link data rate of the satellite, a high gain antenna with a low profile, light weight, and small volume, in addition to a cheap assembly process, is required. These prerequisites can be obtained by utilizing a reflectarray antenna, which comprises a spatial feed and a planar structure. Reflectarray antennas are based on focusing the incident fields from an antenna feeding element to obtain the required radiation pattern by compensating for the phase differences between the reflectarray elements. There are different types of reflectarrays, such as planar microstrip reflectarrays [2,3,4,5,6] and dielectric resonator reflectarrays [7,8,9]. The main feature of the reflectarray is that its radiation characteristics can be manipulated by tuning the geometrical dimensions of its unit cells [10,11,12]. 

Although reflectarrays for high-gain pencil beam patterns in a certain direction can be easily designed using analytical equations [13], the synthesis of shaped or contoured beams is a challenging problem. These shaped or contoured beams are required in many satellite communication systems for better power management [14]. In order to generate a specific radiation pattern, different algorithms have been used to synthesize and optimize the phase distribution on the reflectarray elements. The optimization methods used to synthesis the reflectarray pattern are classified into two main categories: local search algorithms, such as the alternating projections method [15,16], and evolutionary algorithms, such as GA [17,18,19,20], particle swarm optimization (PSO) [21], and the semidefinite relaxation technique [22]. These different optimization algorithms vary in terms of their computation complexity and final convergence rate. It is shown in [23] that the evolutional optimization algorithms are capable of better performance and providing more flexible solutions than the classical optimization algorithms. 

Because a reflectarray antenna consists mainly of a large number of reflecting elements, which contribute to the generation of the required radiation pattern, the required optimization algorithm must be computationally efficient to manage a large number of variables. The reflectarray synthesis problem depends only on phase synthesis, which involves only the phase of the reflected field due to each element in the optimization process. The starting point for the optimization process has a significant effect on the convergence rate. As noted in [24], a good initial point is the phase distribution of a broadside pencil beam pattern. In the present work, three different patterns were investigated: a broadside pencil beam, a tilted pencil bean, and a flat-top beam. Conventional pencil beam and tilted pencil beam reflectarray antennas were designed analytically. The flat-top beam reflectarray antenna was designed using GA optimization. This flat-top pattern is important in remote sensing systems and in different communication technologies, such as 2G/3G/LET cellular bands [25,26,27,28]. 

The reflectarray antenna is usually fed by a horn antenna. However, in this paper, a Yagi–Uda antenna array [29] is used as the feeding antenna for the proposed reflectarray. This Yagi–Uda antenna is characterized by a lower profile and less weight compared to a standard horn antenna [30,31]. Commonly, the incident field distribution on the surface of a reflectarray is approximately represented by an ideal feed model $\cos^{q}\theta$ [32]. The value of the power factor “*q*” is determined by the directivity of the feeding antenna. In the present paper, a more accurate approach is used based on determining the exact field distribution due to the feeding element at the plane of the reflectarray, by separately simulating the feeding element and obtaining its corresponding feed distribution in the required plane. 

By comparison, the reflecting unit cell of the proposed reflectarray antenna is assumed to consist of two stacked circular patches backed by a ground plane. The unit cell is simulated as a periodic structure using Floquet modes [33,34]. Varying the dimension of the unit cell changes the corresponding equivalent surface impedance boundary and therefore the reflection phase shift. The advantage of using this stacked configuration is that it allows the acquisition of a wide range of nearly linear phase-shift changes of more than 360°. This property is important for the implementation of any required phase distribution along the designed reflectarray. 

In this study, GA was chosen to optimize the phases of the reflecting elements to obtain a flat-top radiation pattern with a beam width of around 29°, side lobe level (SLL) of less than −20 dB, and allowable ripple level (ARL) of around −3 dB. The contribution of this work is the optimization of these phases to achieve the desired performance of an optimized flat-top pattern for an MEO system.

This paper is organized as follows. Section 2 introduces the complete reflectarray antenna design, specifications, and modeling procedure. Section 3 presents the analysis of the feeding antenna. In Section 4, the incident field distribution on the plane of the reflectarray, and the corresponding reflection phase distribution for both pencil beam and tilted beam radiation patterns, are obtained. Section 5 introduces the procedures for obtaining the reflection phase distribution for a reflectarray with a flat-top radiation pattern using GA. Section 6 discusses the analysis of the unit cell of the reflectarray. Section 7 presents the results and discussions of three reflectarray designs: broadside pencil beam, tilted pencil beam, and flat-top radiation pattern. The reflectarray of the flat-top radiation pattern was fabricated and measured to show the experimental verification. Finally, Section 8 presents the conclusion.

## 2. Reflectarray Antenna

The proposed reflectarray antenna is composed of an array of reflecting elements arranged on planar circular disk in front of a Yagi–Uda feeding antenna, as shown in Figure 1. It is designed to be operating at a center frequency of 8.5 GHz, which is suitable for the down-link for remote sensing satellite systems. The center of the array is placed at the origin. An x-polarized Yagi–Uda feeder is centered at $\left(x_{f},\;y_{f},\;z_{f}\right)=\left(0,\;0,\;F\right)$, where F is the focal distance of the proposed reflectarray. The focal to diameter ratio of the proposed reflectarray antenna is F/D=1. The diameter of the proposed reflectarray antenna is 352.9 mm which corresponds to $10\lambda_{0}$ at the center frequency. The array elements of the reflectarray are distributed periodically on a square grid of length 17.6 mm, which corresponds to $\lambda_{0}/2$ at the center frequency. The incident field on each reflecting elements at a certain angle can be locally considered as a plane wave with a phase proportional to the distance from the phase center of the feeder to each element. In order to produce a focused beam, the field must be reflected from each unit cell with an appropriate phase shift. This phase shift is adjusted independently for each element to produce a progressive phase shift distribution of the reflected field that produces a focused beam in the required direction.

For a planar array having M×N elements arranged on a rectangular grid on the x−y plane with a uniform separation, the array factor AF(θ,φ) can be written as [15]:(1)$$AF\left(\theta ,\varphi \right)=\overset{M-1}{\underset{m=0}{\sum }}\overset{N-1}{\underset{n=0}{\sum }}A_{m,n}e^{jk\left(md_{x}u+nd_{y}v\right)}$$
where $A_{m,n}$ is the complex excitation of the element (m,n), k is the free space wavenumber, $u=\sin\theta \cos\varphi +\beta_{x}$ and $v=\sin\theta \sin\varphi +\beta_{y}$, and $\beta_{x}\;$ and $\beta_{y}$ are the progressive phase shift between array elements in the x and y directions, respectively. For the case of a pencil beam oriented in the direction $\left(\theta_{o},\varphi_{o}\right)$, the values of these progressive phase shifts can be expressed as:(2)$$\beta_{x}=-\sin\theta_{o}\cos\varphi_{o}\;$$
(3)$$\beta_{y}=-\sin\theta_{o}\sin\varphi_{o}\;$$
where $d_{x}\;$ and $d_{y}$ are the spacing between each two successive elements in the x and y directions, respectively. Thus, the phase shift on the *mn*th element $\Delta \varphi_{m,n}$ is obtained as $\Delta \varphi_{m,n}=\beta_{x}+\beta_{y}$. For the case of a reflectarray, the magnitude of the excitation at the elements $\left|A_{m,n}\right|$ is determined by the amplitude distribution of the fields due to the feeding element on the aperture of the reflectarray. On the other hand, the total phase at the *mn*th element is the summation of the phase of the field distribution of the feeding element plus the phase distribution of the reflection coefficient on the aperture of the reflectarray. The key point in the design of the reflectarray antenna is to determine the required total phase distribution for the obtained amplitude distribution by the feeding element to obtain the required radiation pattern. The next step is the implementation of the reflecting elements of the reflectarray antenna to verify this total phase distribution. For simple radiation patterns such as a broadside pencil beam or a tilted pencil beam, this phase distribution can be obtained analytically in a closed form as follows:(4)$$\Delta \varphi_{m,n}=2\pi N^{\prime }-k\left(r_{m,n}-\vec{R}\cdot \hat{r_{o}}\right)$$
where $N^{\prime }=1,2,3,\ldots$, $r_{m,n}$ is the distance from the feed to each array element, $\hat{R}$ is the position vector from each element to the array center (0,0,0) and $\hat{r_{o}}$ is the position vector in the direction of the main beam of the reflectarray. However, for reflectarray antennas having beams with more complicated shapes, this phase distribution is obtained using optimization techniques.

## 3. Feeding Antenna

In this section, the analysis and design of the feeding antenna for the proposed reflectarray are discussed. The proposed feeding antenna is a Yagi–Uda antenna, as shown in Figure 2a. It consists of a fed dipole antenna inserted between two parasitic elements: a director and a reflector element. The reflector and the director elements enhance the radiation in the direction of the aperture of the reflectarray. Typically, the feed element length $L_{2}$ is usually around 0.45–0.49λ, while the director length $L_{3}$ is approximately 0.4λ to 0.45λ. In addition, the reflector length $L_{1}$ is slightly greater than the fed element. The separation between the elements $d_{s}$ is found to be around 0.1 λ. The radius of these wire elements (a) is set to around approximately 0.025 λ. The proposed Yagi–Uda antenna is simulated using HFSS. Parametric studies are performed through EM simulation for setting the optimum values of the dimensional parameters of an X-band Yagi–Uda antenna at the operating frequency of 8.5 GHz. The required performance of the feeding antenna of the reflectarray in the present case comprises input matching below −10 dB and forward to backward radiation of more than 10 dB. The optimum values of the proposed Yagi–Uda antenna are $L_{1}=20.9\;\mathrm{mm}$, $L_{2}=16.5\;\mathrm{mm}$, $L_{3}=10.8\;\mathrm{mm}$, $d_{s}=3.9\;\mathrm{mm}$, and a=0.5 mm.

Figure 2b shows the simulated $\left|S_{11}\right|$ for the feed antenna. It can be noted that $\left|S_{11}\right|$ at the required operating frequency 8.5 GHz is less than −15 dB, which represents a good matching. The simulated realized gain pattern is shown in Figure 2c. The peak gain is obtained in the –ve z direction towards the aperture of the reflectarray. The peak gain is greater than 6 dBi and the backward radiation is less than −12 dBi; thus, the front to back ratio is around 18 dB. Thus, the proposed Yagi–Uda antenna is suitable for the proposed requirements for the feeding antenna of the reflectarray.

## 4. Field Distribution of the Feeding Antenna on the Aperture of the Reflectarray

In order to obtain the required beam pattern of the reflectarray, the amplitudes and phase distribution on the plane of the reflectarray should be determined. In previous studies of reflectarray antennas, the field distribution is presented as a simple analytical approximation based on ${\left(\cos\theta \right)}^{q}$, where the value of q is chosen to obtain the corresponding approximate radiation pattern of the feeding antenna. In addition, the phase distribution of the field on the aperture of the reflectarray is calculated in terms of the distance from the center of the feeding point to each point on the aperture of the reflectarray. In this paper, this approximation is replaced by directly calculating this field distribution numerically using the commercially EM simulation software HFSS. The advantage of this method is that it does not require any assumptions. Figure 3a shows the 2D distribution of the magnitude of the complex total field on an aperture located at a distance F=352.9 mm from the center of the fed element of the Yagi–Uda antenna. It should be noted that this is only a calculation plane and it does not represent any additional boundary to the simulation problem. This plane and the feeding antenna are included inside a common radiation boundary in the simulation process. This distribution can be presented as a radial function of the magnitude around the z-axis. Figure 3b shows the 1D distribution of the normalized amplitudes along the x-axis at the plane of the reflectarray. In addition, one can also obtain the corresponding phase distribution using the argument for this complex field, as shown in Figure 3b. These amplitude and phase distributions are discretized along the proposed grid of the reflectarray to obtain the amplitude and phase of the incident field on each element of the reflectarray.

The next step is to use this amplitude distribution to find the required phase on each element to obtain the required radiation pattern. Then, it is required to design each element to introduce a phase reflection added to the phase distribution of the feeding element, such that the total phase on this element equals the required phase, to obtain the required radiation pattern.

For the case of a pencil beam radiation pattern, the required total phase distribution on the aperture of the reflectarray antenna can be obtained analytically, as shown in Equation (4). Thus, the reflection phase of each reflecting element can be obtained by subtracting the phase of the feeding element from the required phase distribution of the aperture of the reflectarray. For the present reflectarray structure mentioned in Section 2 and the proposed Yagi–Uda feeding element, the required reflection phases on the aperture of the reflectarray for both broadside and tilted pencil beams with a tilting angle 15° are shown in Figure 4a,b, respectively. It should be noted that the total number of reflecting elements is equal to 316  unit cells, arranged uniformly in a planar grid with M×N=20×20 elements.

By comparison, for the case of a shaped beam, such as a flat-top beam, it is required to determine the required phase distribution using an optimization algorithm because it cannot be determined directly using Equations (1)–(4), as in the case of broadside or tilted pencil beams. However, the phase distribution of the broadside pencil beam can be considered as a good starting point for the proposed optimization process to obtain the corresponding phase distribution for the flat-top beam.

## 5. Flat-Top Pattern Synthesized Using a Genetic Algorithm

In order to start the optimization process, the requirements of the flat-top pattern that can be applied in the optimization procedure should be first introduced. To obtain a nearly constant communication link along the coverage angle, an antenna with a flat-top pattern is required. The maximum coverage angle $\theta_{max}$ is defined from the Earth-satellite geometry shown in Figure 5, as:(5)$$\theta_{max}=\cos^{-1}\left(\frac{d_{max}}{h_{o}+R_{e}}\right)$$
where $R_{e}$ is the radius of the Earth and $h_{o}$ is the vertical distance from the satellite to the Earth’s surface. $d_{max}$ is maximum distance from the satellite to the Earth given by the trigonometric equation as:(6)$$d_{max}=\sqrt{{\left(R_{e}+h_{o}\right)}^{2}-R_{e}{}^{2}}$$

The constraints on the required radiation patterns are considered by using appropriate masks. The requirement of the flat-top normalized pattern is given by means of two mask templates, as shown in Figure 6, which impose the minimum and maximum values that the far field must achieve. Thus, if $AF{\left(\theta_{i}\right)}_{\mathrm{dB}}$ is the normalized array factor of the flat-top beam in dB, it should fulfil the following:(7)$$Mask_{l}{\left(\theta_{i}\right)}_{\mathrm{dB}}\leq \;AF{\left(\theta_{i}\right)}_{\mathrm{dB}}\leq Mask_{u}{\left(\theta_{i}\right)}_{\mathrm{dB}}$$
where the upper mask shapes the normalized radiation pattern in the angular span $-\theta_{max}\leq \theta_{i}\leq \theta_{max}$ with an amplitude of 0 dB. This span is assumed as the transition region of the satellite. For the other θ directions, outside of the main beam the SLL limit is assumed to be below −20 dB. The definition of the upper mask of the flat-top beam pattern, as illustrated in Figure 6, is given in dB as follows:(8)$$Mask_{u}{\left(\theta_{i}\right)}_{\mathrm{dB}}=\{\begin{array}{c} \begin{array}{cc} SLL & -90^{o}\leq \theta_{i}\leq -\theta_{max} \end{array}\\ \begin{array}{cc} \;0 & -\theta_{max}\leq \theta_{i}\leq \theta_{max} \end{array}\\ \begin{array}{cc} SLL & \theta_{max}\leq \theta_{i}\leq 90^{o} \end{array} \end{array}$$

The lower mask is mostly used to control the allowable ripple level (ARL) of the shaped beam, which is −3 dB for the flat-top beam. The lower mask in dB is given as:(9)$$Mask_{l}{\left(\theta_{i}\right)}_{\mathrm{dB}}=\{\begin{array}{c} \begin{array}{cc} <-30\;\;\;\;\;\;\;\;\;\;\;\;\;\;\;\;\;\;\;\;\;\;\;\;\;\; & \theta_{i}\geq \theta_{max}-\Delta \theta \end{array}\\ \begin{array}{cc} \;ARL & -\theta_{max}+\Delta \theta \leq \theta_{i}\leq \theta_{max}-\Delta \theta \end{array}\\ \begin{array}{cc} <-30\;\;\;\;\;\;\;\;\;\;\; & \;\;\;\;\;\;\;\;\;\;\;\;\;\;\;\;\;\theta_{i}\leq -\theta_{max}+\Delta \theta \end{array} \end{array}$$where Δθ is the allowable angle between the upper and the lower mask.

To achieve the flat-top radiation pattern, the phase distribution on the reflectarray elements should be determined through an optimization of a properly defined cost function. The cost function is defined as the error between the obtained normalized array factor AF and the required upper and lower masks. The cost function is normalized for $N_{total}$ angles for both the upper and lower limits of the mask as follows:(10)$$cost=\frac{\varepsilon_{u}\left(\theta_{i}\right)+\varepsilon_{l}\left(\theta_{i}\right)}{N_{total}}$$
where $\varepsilon_{u}\left(\theta_{i}\right)$ and $\varepsilon_{l}\left(\theta_{i}\right)$ are given by Equations (11) and (12), respectively:(11)$$\varepsilon_{u}\left(\theta_{i}\right)=\overset{N_{total}}{\underset{i}{\sum }}\left[AF{\left(\theta_{i}\right)}_{\mathrm{dB}}-Mask_{u}{\left(\theta_{i}\right)}_{\mathrm{dB}}\right]\frac{\left[1+\mathrm{sgn}\left(AF{\left(\theta_{i}\right)}_{\mathrm{dB}}-Mask_{u}{\left(\theta_{i}\right)}_{\mathrm{dB}}\right)\right]}{2}$$
(12)$$\varepsilon_{l}\left(\theta_{i}\right)=\overset{N_{total}}{\underset{i}{\sum }}\left[Mask_{l}{\left(\theta_{i}\right)}_{\mathrm{dB}}-AF{\left(\theta_{i}\right)}_{\mathrm{dB}}\right]\frac{\left[1+\mathrm{sgn}\left(Mask_{l}{\left(\theta_{i}\right)}_{\mathrm{dB}}-AF{\left(\theta_{i}\right)}_{\mathrm{dB}}\right)\right]}{2}$$
where $AF{\left(\theta_{i}\right)}_{\mathrm{dB}}=20\log\left(\left|AF\left(\theta_{i}\right)\right|\right)$ and sgn(x)=1 for x>0 and sgn(x)=−1 for x<0. Thus, a better match between the obtained pattern and the required pattern is obtained for the minimum value of this cost function. 

After determining the required radiation pattern for the proposed MEO satellite communication system and the amplitude distribution of the fields due to the feeding element, the corresponding phase distribution on the aperture of the reflectarray must be determined. This phase distribution is obtained using GA. In GA, the optimization starts with an initial population comprising a number of candidate solutions (designated as chromosomes). These parents are controlled using different factors (combination, crossover, or mutation) to make a new set of chromosomes for the next generation. During the advancement of the arrangement, chromosomes are reviewed with respect to the enhancement of the fitness between the obtained radiation pattern and the required mask. The higher-positioned chromosomes are chosen to proceed to the next generation. Once the new generation is formed, the fitness of its chromosomes is estimated and the process continues until the convergence condition is satisfied. The algorithm stops when the value of the fitness function for the best point in the current population is less than or equal to the fitness limit. The important basic genetic algorithm steps are presented in Figure 7.

The main problem when applying GA is the large number of optimization variables, which correspond to all reflecting elements on the reflectarray. This large number of variables requires a large computational time, which affects the overall convergence of the optimization process. However, because the proposed flat-top radiation pattern and amplitude distribution of the fields due to the feeding element are radially symmetric around the z-axis, as shown in Figure 3, the required phase distribution should be also be radially symmetric around the z-axis. Thus, the number of the unknown variables can be reduced by taking into consideration this symmetry. For the case of a circular reflectarray as shown in Figure 1, the reflecting elements can be arranged into four image-symmetric quarters. Each quarter can also be divided into two symmetric halves, such that each column in the quarter would be the same as the corresponding perpendicular raw in the same quarter as shown in Figure 8. 

Using this approach, it is possible to reduce the number of unknowns in the optimization process to one-eighth of the number of reflectarray elements. This significant reduction reduces the convergence computational time and also improves the resulting convergence. Figure 9 shows the phase distribution obtained by using GA to obtain the required flat-top radiation pattern. Moreover, the obtained flat-top radiation pattern is shown in Figure 10. It can be noted that the obtained flat-top radiation pattern almost satisfies the required conditions of the maximum coverage angle of 29°, Δθ of 5°, and ARL of −3 dB.

## 6. Design of the Unit Cell of the Reflectarray

The previous section showed how to obtain the required phase distribution of the reflected fields along the surface of the reflectarray to obtain the required radiation patterns. The following step is to design this reflecting element and to show how it can be controlled to obtain the required phases. The proposed unit cell is composed of two conducting elements of circular shape stacked in two layers of FR4 dielectric slabs with a dielectric constant $\epsilon_{r}=4.4$, as shown in Figure 11. The top substrate has a height $h_{T}=3\;\mathrm{mm}$ and the bottom has a height $\;h_{B}=1.5\;\mathrm{mm}$. The stacked patches are backed with a ground plane. The bottom circular patch has a diameter $\;d_{B}$, whereas the top patch diameter $\;d_{T}=0.75\;d_{B}$. The unit cell has dimensions $d_{x}=d_{y}=17.65\;\mathrm{mm}$. 

The phase of the reflection coefficient of the unit cell as a function of the diameter of the lower circular patch is shown in Figure 12. It should be noted that the diameter of the upper patch depends on the corresponding diameter of the lower patch. It can be noted that the phase of the reflected field can be controlled over a range from 0° to −500° by changing $d_{B}$ from 5 to 14 mm. The reason for using a stacked structure is that it is not possible to obtain such a wide range of phase using a single layer structure. This wide range of phase is suitable for obtaining the required phase distribution for the different cases of the proposed reflectarray antennas. 

## 7. Results and Discussion

In this section, the above analyses for the required phase distributions and the reflecting element are combined to introduce the complete design of the proposed reflectarray antennas. Three designs are presented. The first has a pencil beam with a broadside radiation based on the phase distribution in Figure 4a; the simulation layout of the reflecting elements in this case is shown in Figure 13. It should be noted that directly allocating the dimensions for all these elements in this configuration on a simulation tool such as HFSS is complicated. However, this problem is simplified by generating a lookup table to convert the phase at each point on the reflectarray plane to the corresponding radii for the upper and lower circular patches of the corresponding element. Then, these radii with the corresponding centers are formatted as a Visual Basic Script (VBS), which is loaded directly by HFSS to draw the reflecting elements. This procedure introduces a significant improvement in developing the simulations. Figure 14 shows the simulated 3D radiation pattern of a broadside pencil beam with peak gain around 22 dBi. In Figure 15, the simulation layout of the reflecting elements for the tilted beam radiation pattern based on the phase distribution in Figure 4b is shown. The simulated 3D radiation pattern in this case is presented in Figure 16 and the peak gain is found to be nearly 21 dBi. Figure 17a shows the simulated layout of the reflecting elements for the flat-top pattern. This configuration was fabricated and measured to validate the flat-top pattern obtained by GA and that obtained by numerical calculation. Figure 17b–d shows the fabricated layers of the flat-top reflectarray antenna.

Moreover, a Yagi–Uda antenna was fabricated, as shown in Figure 18a, to complete the structure of the reflectarray. The frequency response of the reflection coefficient magnitude of the fabricated Yagi–Uda antenna was measured using a vector network analyzer (VNA; Rhode and Schwartz model ZVA67), as shown in Figure 18b. The excellent matching of the fabricated antenna at the operating frequency of 8.5 GHz can be noted in Figure 18c. 

Figure 19 shows the complete reflectarray antenna with its feeding antenna. The radiation patterns of the fabricated antenna were measured inside as anechoic chamber, as shown in Figure 20. Figure 21 shows the measured normalized radiation pattern of this reflectarray antenna compared to the radiation pattern obtained by GA for the required phase distribution. It can be noted that the obtained radiation pattern satisfies the required mask to a good extent. The slight differences in the obtained radiation pattern can be explained due to the alignment and the fabrication accuracy. In addition, Figure 22 shows the 3D radiation pattern of the flat-top beam with peak gain of around 8 dBi.

## 8. Conclusions

This paper presents the analysis and design of a flat-top reflectarray antenna for an MEO satellite system for remote sensing at an X-band frequency of 8.5 GHz. The feeding antenna is a Yagi–Uda antenna. The amplitude and phase of the field distribution due to the feeding antenna at the aperture of the reflectarray antenna are obtained numerically without introducing any approximation. This field distribution is used to obtain the required reflection phase distribution to obtain the required flat-top radiation pattern. This phase distribution is obtained using genetic algorithm optimization. The initial phase distribution of the optimization process is taken to be the phase distribution of a broadside pencil beam, which is obtained analytically. The problem of the tilted pencil beam is also investigated using analytical calculations. These reflection phase distributions are implemented using periodic reflecting elements. The reflecting element is composed of stacked circular patches on a grounded double-layered dielectric substrate. The reflection phases of these reflecting elements are adjusted by controlling the diameter of the stacked circular patches. These reflecting elements are arranged according to the required phase distribution for the cases of broadside pencil beam, tilted pencil beam, and flat-top beam. The complete reflectarray systems for the three cases are investigated. Details of simulation steps are discussed. The flat-top beam reflectarray was fabricated and measured for verification. Good agreements between the obtained results and the simulated results were obtained. The results of the reflection coefficient of the designed and fabricated feeding antenna showed a good agreement. Moreover, the obtained radiation pattern of the complete reflectarray antenna was found to satisfy the required radiation mask to a good extent. 

## Figures and Tables

**Figure 1 sensors-22-01166-f001:**
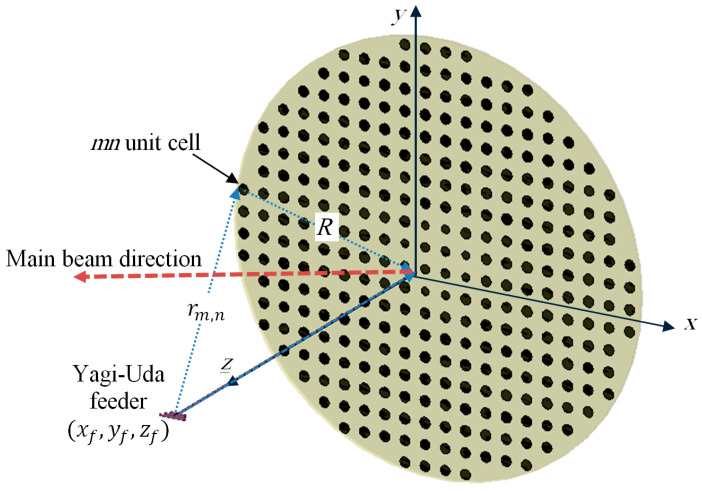
A reflectarray antenna fed by a Yagi–Uda antenna.

**Figure 2 sensors-22-01166-f002:**
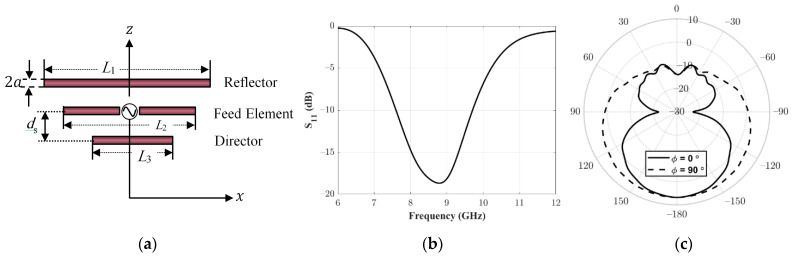
Proposed Yagi–Uda feeding antenna. (**a**) Geometry, (**b**) Simulated Reflection coefficient, (**c**) Simulated total gain pattern.

**Figure 3 sensors-22-01166-f003:**
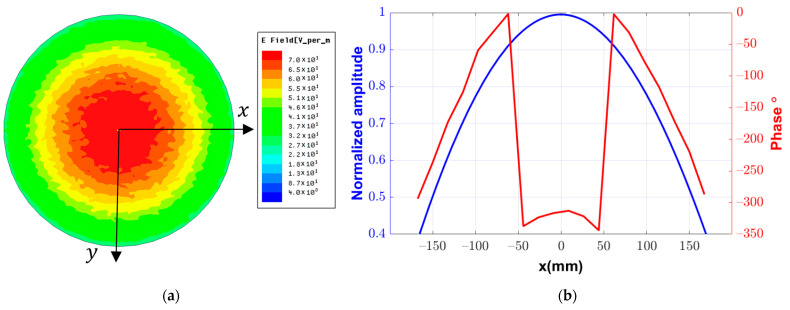
Complex field distribution of the feeding element at the plane of the reflectarray: (**a**) 2D representation, (**b**) normalized amplitude and phase along the x-axis.

**Figure 4 sensors-22-01166-f004:**
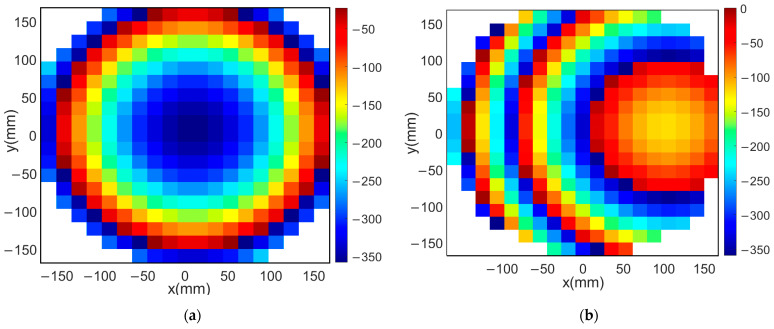
Phase distribution of the pencil beam reflectarray: (**a**) broadside beam, (**b**) tilted beam with a tilting angle of 15°.

**Figure 5 sensors-22-01166-f005:**
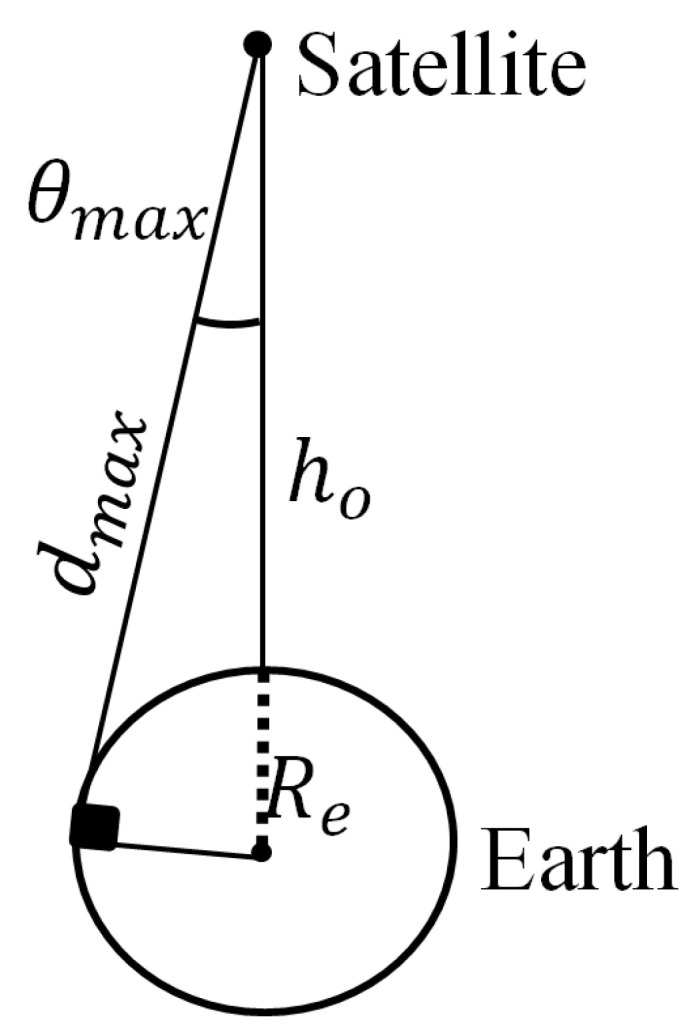
Geometry of the satellite coverage of the MEO satellite system.

**Figure 6 sensors-22-01166-f006:**
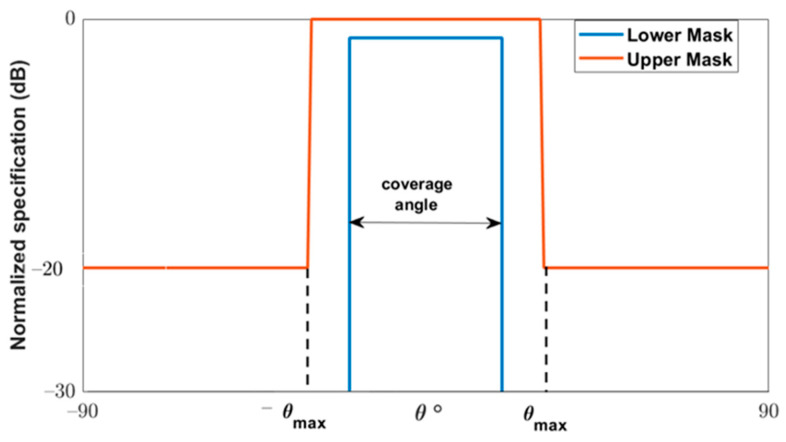
Required mask for the normalized radiation pattern.

**Figure 7 sensors-22-01166-f007:**
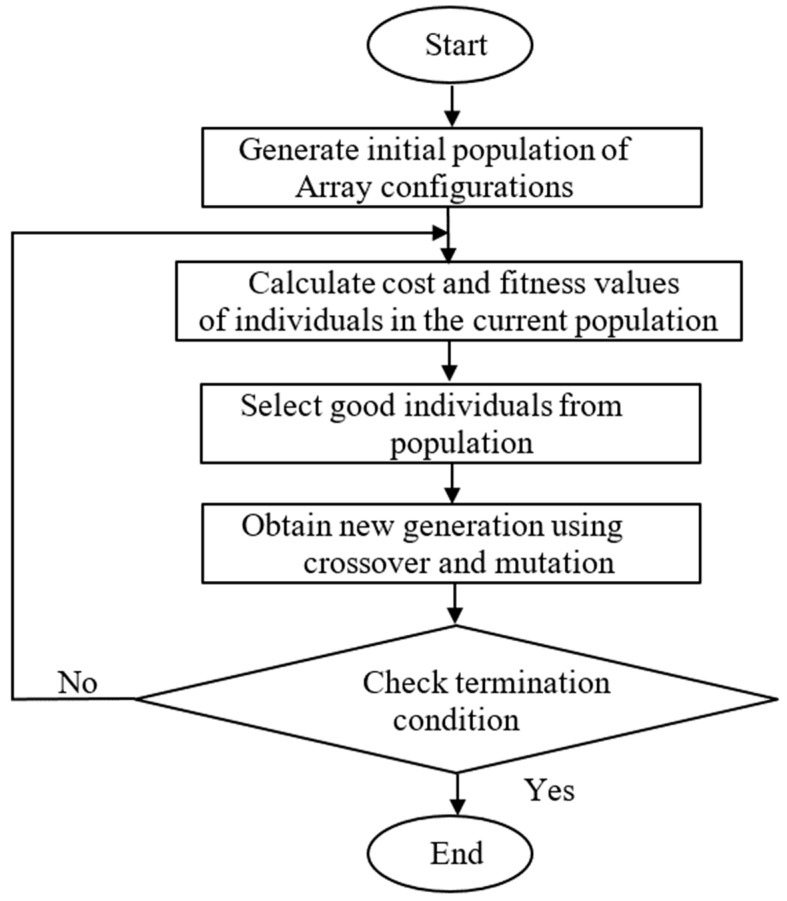
Flowchart of a general GA approach.

**Figure 8 sensors-22-01166-f008:**
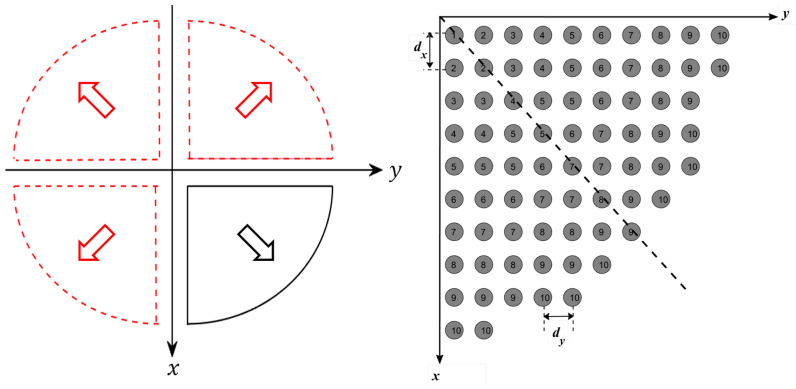
Phase symmetry on half of the elements of the proposed reflectarray antenna.

**Figure 9 sensors-22-01166-f009:**
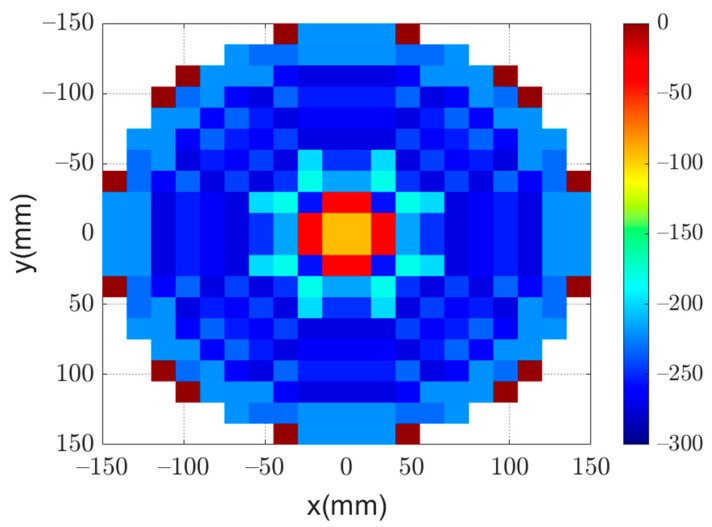
Optimized phase distribution of a flat-top radiation pattern.

**Figure 10 sensors-22-01166-f010:**
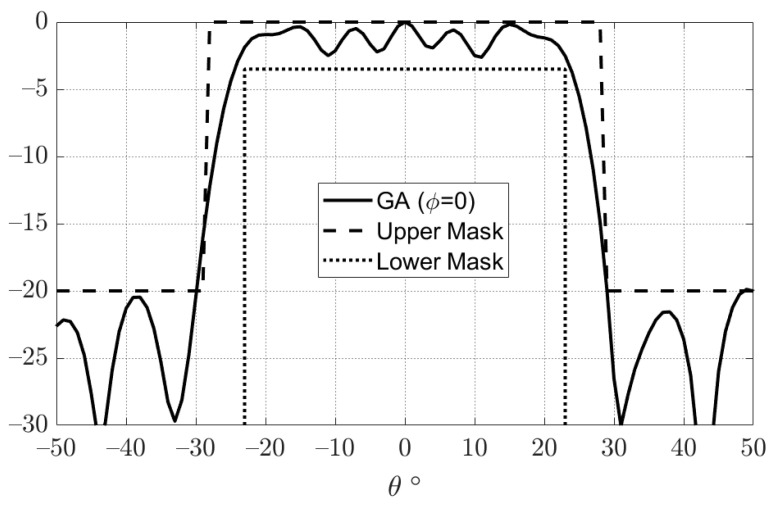
Flat-top radiation pattern obtained using a genetic algorithm.

**Figure 11 sensors-22-01166-f011:**
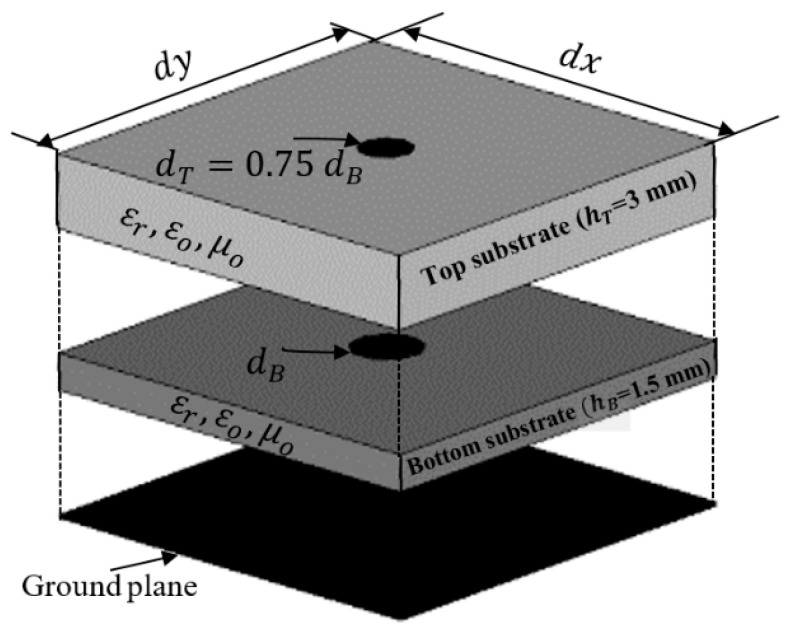
Geometry of the proposed unit cell.

**Figure 12 sensors-22-01166-f012:**
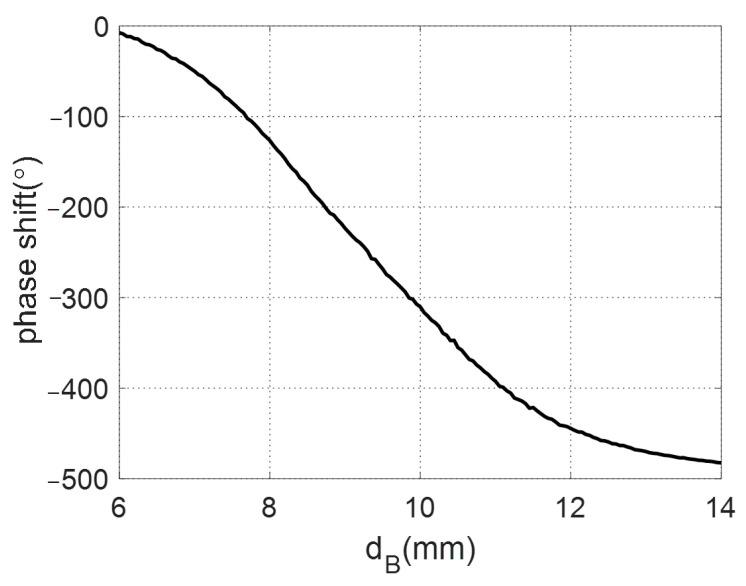
Reflection phase response of the unit cell.

**Figure 13 sensors-22-01166-f013:**
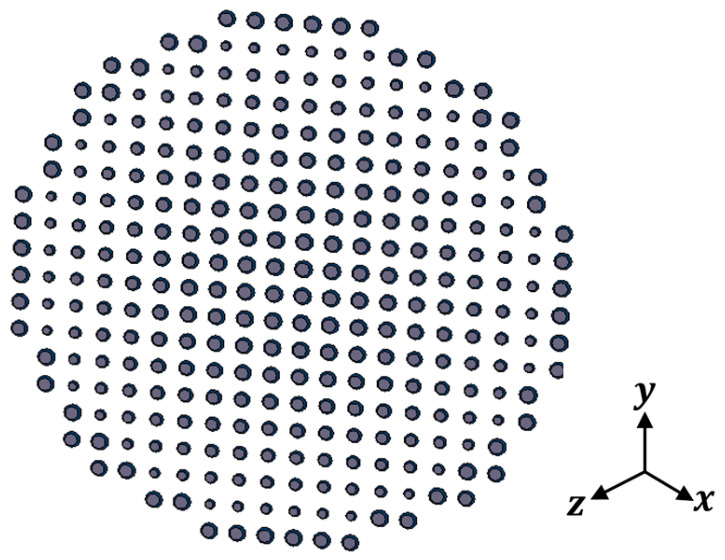
Simulation layout of the reflecting elements for the broadside pencil beam pattern.

**Figure 14 sensors-22-01166-f014:**
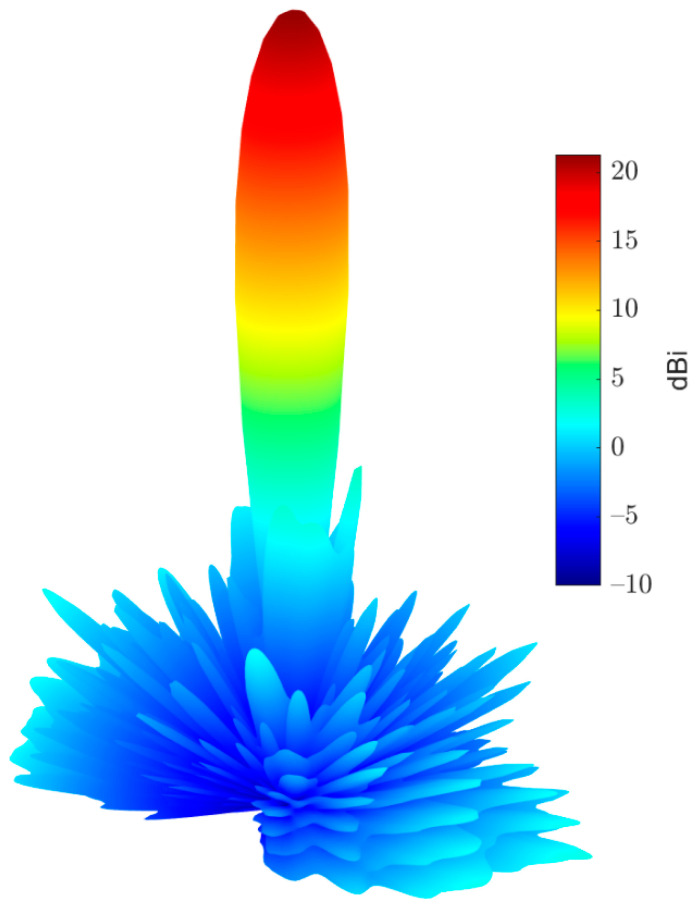
3D radiation pattern of the simulated broadside pencil beam reflectarray antenna.

**Figure 15 sensors-22-01166-f015:**
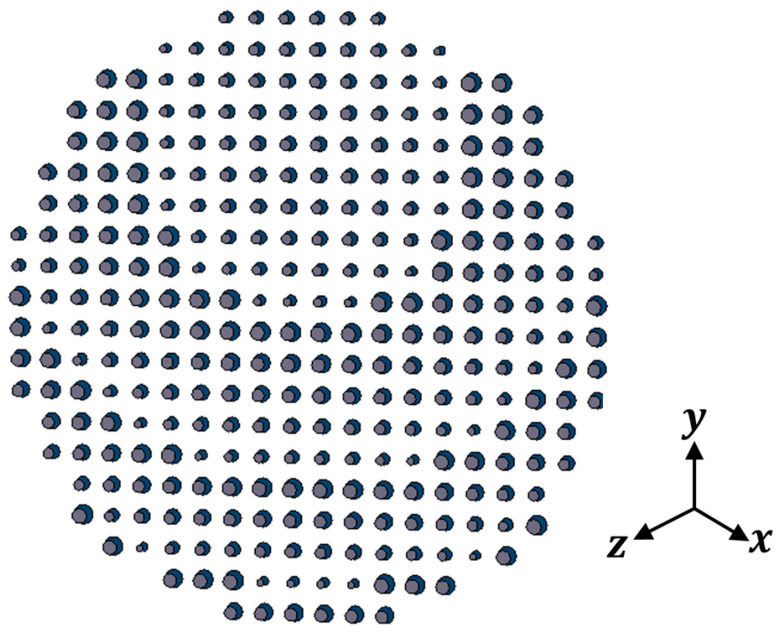
Simulation layout of the reflecting elements for the tilted beam pattern.

**Figure 16 sensors-22-01166-f016:**
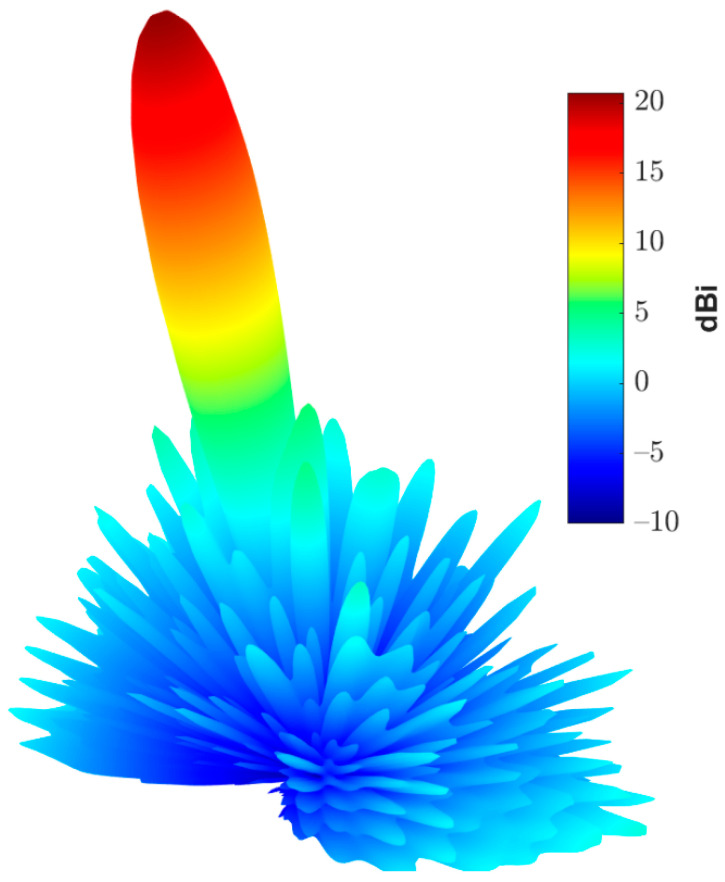
3D radiation pattern of the simulated tilted beam reflectarray antenna.

**Figure 17 sensors-22-01166-f017:**
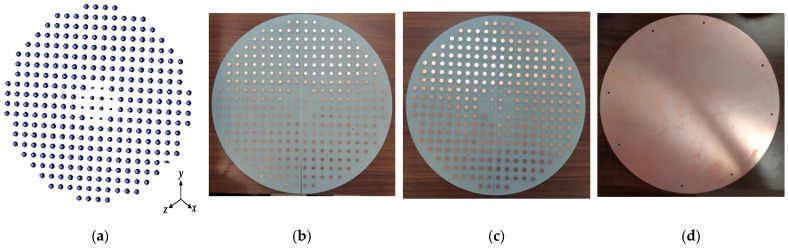
Reflecting elements for the flat-top pattern: (**a**) simulated layout, (**b**) fabricated upper layer, (**c**) fabricated bottom layer, and (**d**) fabricated ground plane.

**Figure 18 sensors-22-01166-f018:**
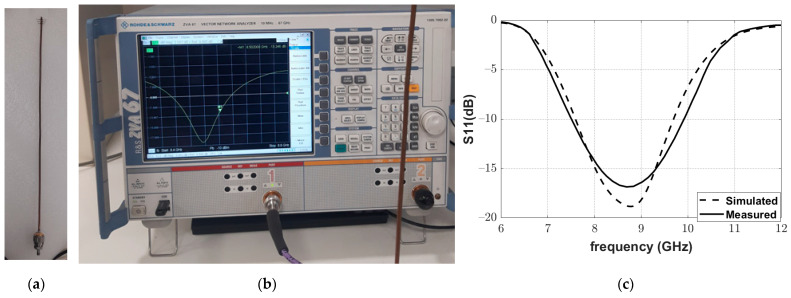
Measurement of the reflection coefficient of the fabricated Yagi–Uda antenna: (**a**) the fabricated Yagi–Uda antenna, (**b**) Yagi–Uda antenna connected to the Rhode and Schwartz model ZVA67 VNA, and (**c**) simulated and measured magnitude of the reflection coefficient.

**Figure 19 sensors-22-01166-f019:**
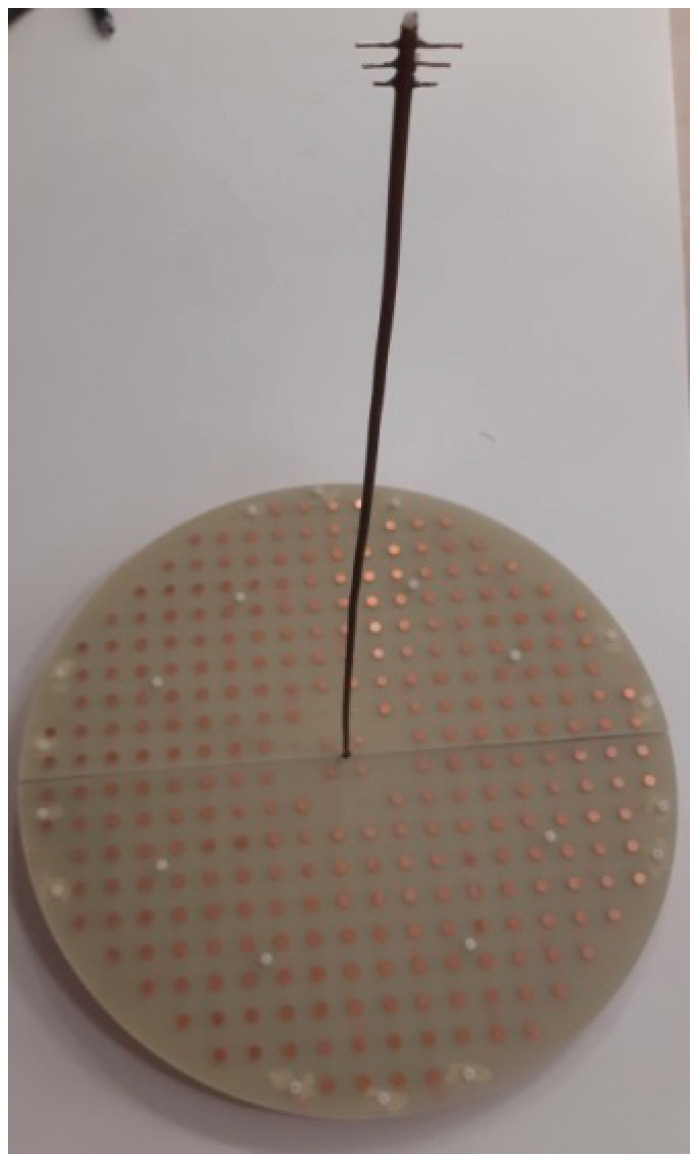
Fabricated prototype of the proposed reflectarray antenna.

**Figure 20 sensors-22-01166-f020:**
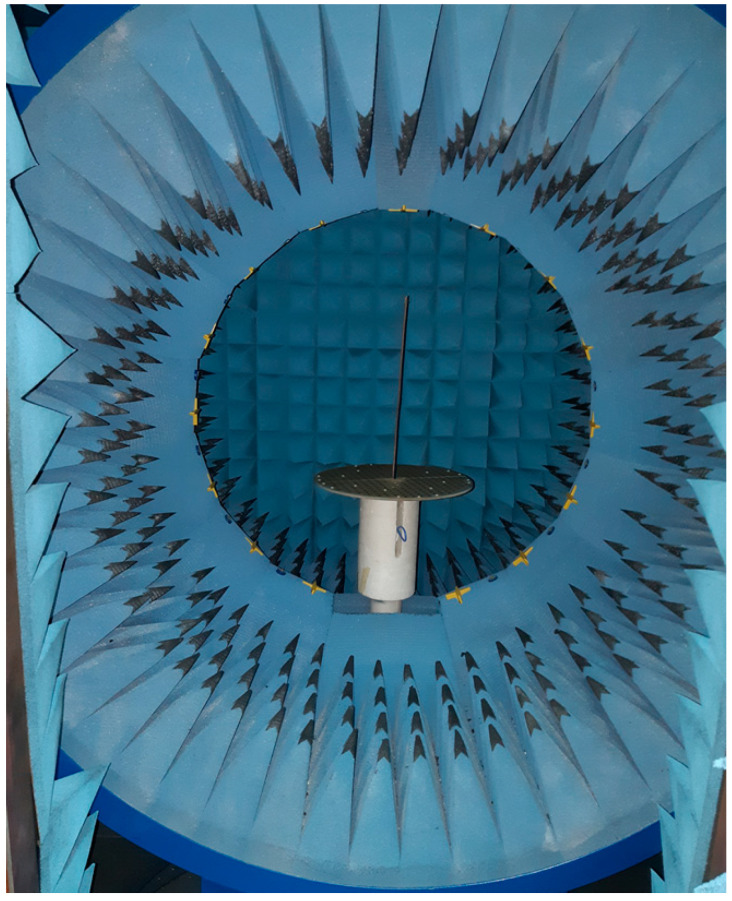
Fabricated antenna inside the anechoic chamber for the radiation pattern measurement.

**Figure 21 sensors-22-01166-f021:**
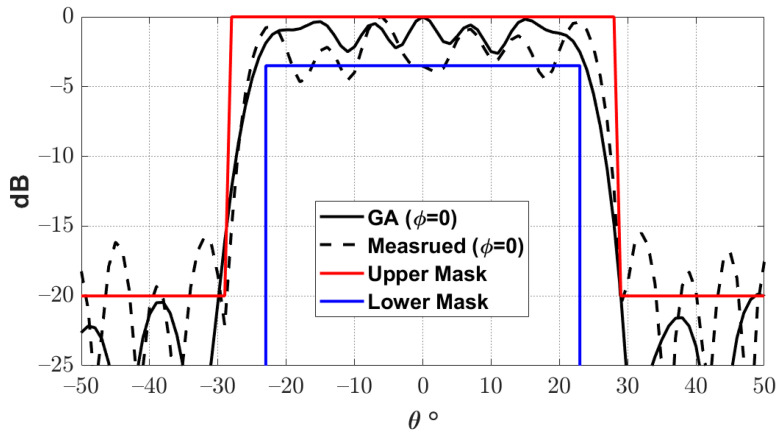
Measured and optimized radiation pattern of the proposed reflectarray antenna.

**Figure 22 sensors-22-01166-f022:**
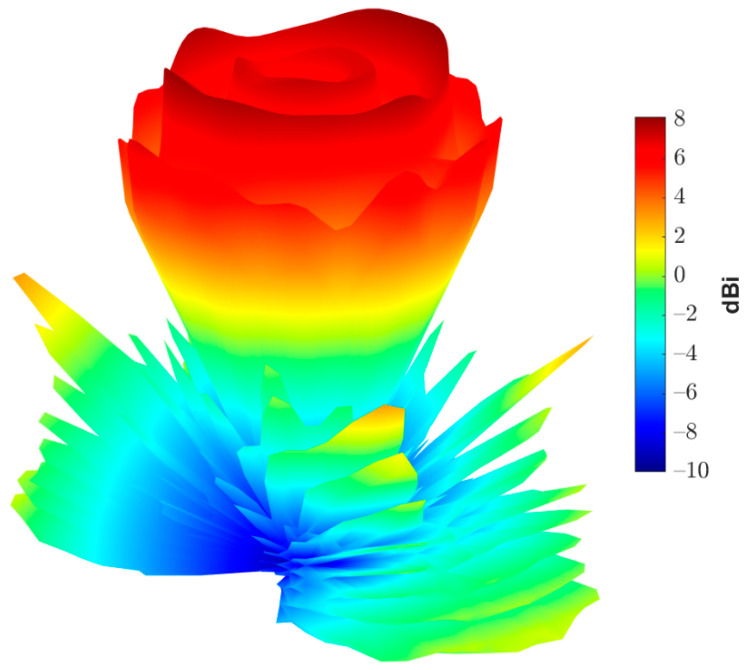
Three-dimensional radiation pattern of the proposed reflectarray antenna.

## Data Availability

Data sharing not applicable.

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
