# Peer review of "Analysis and Design of an X-Band Reflectarray Antenna for Remote Sensing Satellite System"

_sensors, 2022, doi:10.3390/s22031166_

Round 1

Reviewer 1 Report

COMMENTS:

The submitted paper presents the design of reflectarrays for the synthesis of broadside, tilted, and flat-top patterns. The reflectarrays operate at 8.5 GHz focusing on MEO satellites. The topic is of interest to most readers of Sensors, but many parts of the manuscript need improvement before publication. Some comments are listed below to improve the quality of the paper.

1) Please try to improve the resolution of Figures 1 and 2.

2) Please check the expression for v in line 116.

3) Please check if equation (1) is an array factor for an array of MxN elements as stated in line 113.

4) Please check if the phase shifts beta_x and beta_y are multiplied by k in equation (1).

5) Please check equation (4) because it seems to present a dimensional problem. In addition, is the variable N the same as used in equation (1)?

6) Is the distance between the reflector and active dipoles of the Yagi in Figure 2 also equal to ds?

7) The authors should describe the EM technique/software used to carry out the design of the Yagi-Uda antenna as well as the criteria employed in the design (for example, return loss, gain, front-to-back ratio, etc.).

8) Please check “is obtained in –ve z direction” in line 155.

9) Why the maximum normalized amplitude in Figure 3(b) is not equal to 1?

10) The phase in Figure 3(b) is correspondent to the x component of the electric field?

11) The authors should clarify how the graphs in Figure 4 were computed. They state in line 203 they used an analytical technique, but it is not clear which one was employed.

12) Can the constant power illumination mentioned in line 210 be obtained with a flat-top or cosecant-squared pattern? A better explanation should be given to justify the flat-top pattern.

13) Please check the legend of Figure 6. The colors of upper and lower masks seem to be changed.

14) Check the lower mask defined in equation (9). It is not clear the value 10^-6 in comparison to Figure 6.

15) The functions (11) and (12) used to compute the cost function have the array factor, which can be a complex number. Is there any advantage to this choice? The use of the absolute value of the array factor, which is directly related to the radiation pattern, could be an option?

16) Lines 204 to 206 state that the phase distribution of the broadside pencil beam can be considered as a good starting point for the proposed optimization process. It is not clear how can this phase distribution be used in a Genetic Algorithm. Is this phase distribution set as a chromosome of the first generation?

17) Please try to improve the resolution of Figure 11.

18) The computation of Figure 12 must be explained in more detail. For example, it was not mentioned if the plane wave was incident perpendicular to the unit cell.

19) Instead of showing Figure 18, it is better to present a graph containing the simulated and measured reflection coefficient magnitudes. The picture of the network analyzer screen is not appropriate to read numerical data.

20) Why does the normalized measured pattern in Figure 21 not reach the 0 dB value? Is it a pattern for the total field or a specific component (theta, phi)?

21) 21) It is not clear whether Figure 22 shows a simulated or measured pattern (since StarLab was used)?

22) The literature review in the Introduction must be expanded to explain the new contributions brought by the paper. If possible, try to present a Table to compare the features of the proposed design with those found in previous references.

Reviewer 2 Report

See the attachment please

Reviewer 3 Report

  1. It is not clear how does GA contribute to antenna design. Authors should discuss how did authors applied GA.
  2. Fig.21 shows final result of the paper but ripple level is too high. Please discuss errors between the measurement and simulation results, too.
  3. Radiation pattern of the designed reflectory antenna is theta = 0 deg where the feed antenna exists. It is clear that the interaction between the reflected wave and the feeding antenna is significant. This point should be discussed.

Round 2

Reviewer 1 Report

The suggestions proposed by this reviewer to the authors were all applied to the manuscript. Thanks.

Reviewer 2 Report

The authors have considered all comments and revised the manuscript carefully.

The introduction is acceptable now with explaining more relevant and most recent papers.

The paper can be published as it is.

Reviewer 3 Report

No further comments